# Heat Stress in Indoor Environments of Scandinavian Urban Areas: A Literature Review

**DOI:** 10.3390/ijerph16040560

**Published:** 2019-02-15

**Authors:** Karin Lundgren Kownacki, Chuansi Gao, Kalev Kuklane, Aneta Wierzbicka

**Affiliations:** Ergonomics and Aerosol Technology, Department of Design Sciences, Lund University, 221 00 Lund, Sweden; chuansi.gao@design.lth.se (C.G.); kalev.kuklane@design.lth.se (K.K.); aneta.wierzbicka@design.lth.se (A.W.)

**Keywords:** indoor environments, heat stress, climate change, urban heat island, preventive actions

## Abstract

Climate change increases the risks of heat stress, especially in urban areas where urban heat islands can develop. This literature review aims to describe how severe heat can occur and be identified in urban indoor environments, and what actions can be taken on the local scale. There is a connection between the outdoor and the indoor climate in buildings without air conditioning, but the pathways leading to the development of severe heat levels indoors are complex. These depend, for example, on the type of building, window placement, the residential area’s thermal outdoor conditions, and the residents’ influence and behavior. This review shows that only few studies have focused on the thermal environment indoors during heat waves, despite the fact that people commonly spend most of their time indoors and are likely to experience increased heat stress indoors in the future. Among reviewed studies, it was found that the indoor temperature can reach levels 50% higher in °C than the outdoor temperature, which highlights the importance of assessment and remediation of heat indoors. Further, most Heat-Health Warning Systems (HHWS) are based on the outdoor climate only, which can lead to a misleading interpretation of the health effects and associated solutions. In order to identify severe heat, six factors need to be taken into account, including air temperature, heat radiation, humidity, and air movement as well as the physical activity and the clothes worn by the individual. Heat stress can be identified using a heat index that includes these six factors. This paper presents some examples of practical and easy to use heat indices that are relevant for indoor environments as well as models that can be applied in indoor environments at the city level. However, existing indexes are developed for healthy workers and do not account for vulnerable groups, different uses, and daily variations. As a result, this paper highlights the need for the development of a heat index or the adjustment of current thresholds to apply specifically to indoor environments, its different uses, and vulnerable groups. There are several actions that can be taken to reduce heat indoors and thus improve the health and well-being of the population in urban areas. Examples of effective measures to reduce heat stress indoors include the use of shading devices such as blinds and vegetation as well as personal cooling techniques such as the use of fans and cooling vests. Additionally, the integration of innovative Phase Change Materials (PCM) into facades, roofs, floors, and windows can be a promising alternative once no negative health and environmental effects of PCM can be ensured.

## 1. Introduction

Increasing heat exposure levels are one of the most certain effects of climate change [1], and, in the urban context, the urban heat island effect [2,3] contributes to additional heat exposure, thus affecting housing and their indoor environments in a complex way. In regards to Northern Europe, a climatological study by Tomczyk et al. (2017) confirms an increase in both the mean maximum air temperature in the summer and the frequency of the occurrence of relatively warm days during the period between 1973–2010 [4]. The consequential impacts on the indoor environment consists of interactions between the outdoor climate in combination with technology, individual behaviors, and social systems [5]. In 2009, the World Health Organization (WHO) highlighted the lack of knowledge of the development of severe heat indoors [6].

To date, heat and health studies have mainly focused on the outdoor environment, and the relationship between hot outdoor temperatures and mortality is well established. For example, Oudin Åström et al. (2018) found that heat waves significantly increased both all-cause mortality and coronary heart disease mortality by approximately 10–15%, respectively, in the Swedish context [7]. There is especially a need for studies in developed countries, as studies about indoor environments show that the population in these countries spend most of their time—about 90%—indoors [8,9,10,11,12,13,14,15]. The health outcomes of heat exposure depend on the climatic zones specific for given regions [16]. Furthermore, several studies of outdoor conditions during heat waves show that mortality is higher in urban areas compared with the surrounding countryside. One reason for this may be that the urban heat island maintains a high nighttime temperature [17,18].

In general, an overview of the state of knowledge and undertaken studies with regards to heat problems in indoor environments is lacking. This study on heat problems in indoor environments was performed at the request of the Public Health Agency of Sweden due to the lack of knowledge in this area. The study is a part of a larger project at the Public Health Agency of Sweden, funded by the Swedish Contingencies Agency, on how and where severe heat can occur, be identified, and managed both in the outdoor and indoor environments in Sweden. The specific aims for this paper are three-fold:-To investigate how severe heat can develop indoors;-How severe heat can be identified in indoor environments;-To describe local level actions that have scientific support.

Severe heat, as referred to in the paper, occurs when the body’s means of controlling its internal temperature starts to fail. As well as air temperature, factors such as humidity, radiation, wind speed, metabolic rate, and the clothing may contribute to severe heat stress.

## 2. Materials and Methods

To identify previous research and further research needs, the literature review method was applied. The review is limited to existing buildings in the urban areas of Scandinavia and includes schools, retirement homes, apartments, preschools, and non-industrial offices (those without industrial processes that produce heat). A limited systematic review of the literature was conducted using two extensive databases: PubMed and Google Scholar. The search terms used were indoor climate/temperature/thermal comfort/heat stress/exposure/heat wave/climate change/health/solutions/prevention/reduction, personalized cooling, personal cooling/protection, indoor environment quality/indoor environmental quality, housing, energy savings, and homes. The results from the literature search were complemented by the authors’ own collections of relevant literature.

Most of the found literature came from North America or Germany. The Scandinavian literature consisted primarily of studies on thermal comfort, air quality, energy efficiency, passive housing, ventilation, and heating. Overall, there is limited research in relation to heat waves and indoor environments as most studies have focused on the outdoor environment. In general, the literature search resulted in articles that examined air quality, thermal stress in occupational setting, thermal comfort, and productivity studies in office environments, or studies of heat waves in outdoor environments related to human ill health and mortality. Moreover, the majority of thermal comfort research focused on the health effects of low temperatures, and air quality studies have focused on exposure of contaminants and specific disease processes. In recent years, however, efforts have been made to more accurately describe the complex interactions between health and the indoor environment, including heat [8].

## 3. Results and Discussion

### 3.1. The Relationship between the Outdoor and Indoor Climate

In the northern hemisphere, heat-related deaths commonly occur at home. However, little is known of the connections between the outdoor and indoor climate [11]. For buildings without air conditioning, climate parameters in the outdoor environment such as temperature, solar radiation, and humidity may be directly related to indoor conditions; however, these relationships are specific to location, type of building, and season. In buildings with air conditioning, the ratio of outdoor temperature to indoor temperature is not that strong [19].

In general, studies have shown that high outdoor temperatures generally increase indoor temperatures [11,12,13,20,21]. For example, a case report by Hawkins-Bell and Rankin (1994) documented that the indoor temperature could reach up to 18.8 °C higher compared to the outdoor temperature (35.6 °C) during a heat wave in Philadelphia in 1994 [21]. However, the sensitivity between the outdoor and indoor temperature varies depending on the type of building (e.g., type of building material, size and orientation of windows), outdoor temperature, socioeconomic status, and neighborhood [11,14], as well as the diurnal cycle conditions [10] and behavioral factors such as cooking, bathing, and use of air conditioning [11]. Personal heat exposure studies using personal data loggers for temperature and humidity also show that individuals are exposed to a wide range of temperatures during the summer [22,23]. For example, Franck et al. (2013) found that indoor temperatures were related to outdoor temperatures, but that the relationship is complex. Even within a residential area or a building, people can be affected differently, due to, for example, shading, orientation of the apartment, and apartment location within the building. Temperatures also tend to increase with floor number (elevation) and proximity to the center of the urban area (less green spaces). The authors also found that the subjective experience of heat did not clearly reflect the measured temperature differences but varied with the individual’s adaptability [12].

In regards to socioeconomic status, Roberts and Lay (2013) studied a set of 60 North American homes with older adults in three different climates (Florida, Oregon, New York/Washington State). The researchers found that older adults and patients in the neighborhood of more African Americans had warmer and more humid indoor environments than in the other neighborhoods, which provides some support for a link between demographic factors such as poverty and housing environment with exposure to dangerous indoor heat [17]. However, this link is specific to American conditions and less clear in a European context [24]. Roberts and Lay (2013) also found that buildings in residential areas with a higher proportion of single persons tended to be somewhat cooler and drier than neighborhoods with more multi-family homes. This result may be related to heat resulting from crowding or to an increased use of air conditioning at workplaces and well-off urban areas. Furthermore, it was found that the living rooms were generally cooler than the bedrooms [17].

Chan et al. (2001) analyzed the indoor-outdoor relationship using a fictional built environment and a human physiological model to understand important risk factors associated with negative health effects from heat waves. The method consisted of integrated modules—including environmental conditions and behavioral responses—linked to a physiological model that predicted the inhabitant’s core body temperature [25]. The method was applied to the conditions during the Chicago heat wave in 1995 [26], and the results were compared to published results. The authors found a connection between heat exposure and the risk of mortality in individuals, but, when aggregating for populations, the relationship was less clear [25].

In summary, for buildings without air conditioning, which are the most common in Scandinavia, several studies have found a strong correlation between the outdoor and indoor temperature [10,11,12,13,14,15]. Furthermore, the humidity relationship between the indoor and outdoor conditions tends to be stronger than the temperature relationship [17,20,27].

### 3.2. Risk Groups

The development of severe heat indoors is of particular importance to the health of the most vulnerable—in particular children and the elderly [28]. For example, mortality from heat waves has been related to cardiovascular and respiratory diseases, which are more common in the elderly [29]. The European heat wave of 2003 had a serious impact on elderly people in hospitals and residential buildings. Mortality doubled in the age group 75+ in French retirement homes [30]. A study in the UK also found a much higher risk of heat stroke in hospitals’ patients and the lowest risk for people at home [31]. In addition to the natural aging patterns, several medical conditions increase the vulnerability to heat stress [32], as well as a number of medications affect heat regulation [24]. Uejio et al. (2016) examined the indoor climate (temperature and humidity) in homes of individuals who needed emergency medical care in New York City during the summer of 2013. The study found that warmer and sunny outdoor conditions quickly increased the temperature and humidity indoors. In addition, the elderly tended to live in warmer buildings [13,14], which, in part, could be the result of elderly staying in older buildings with poorer ventilation or lack of air conditioning. The elderly may also have difficulties controlling the indoor temperature due to impaired physical and mental ability and be less likely to change their behaviors and habits.

Epidemiological studies show that people with depression, cardiovascular disease, and diabetes must be extra careful in hot weather. People with diseases affecting mobility, awareness, and behavior— such as people with dementia and Parkinson’s disease—also constitute a risk group [24]. It is likely that some identified risk groups, such as elderly people and people with dementia, are more likely to stay indoors under heat waves. Van Loenhout et al. (2016) examined the relationship between indoor temperature and heat-related health problems in elderly people and found that the relationship is stronger indoors than outdoors [33].

In addition to the elderly, chronically ill, people who take certain medications or have a disability, the other identified risk groups are children under 12 months (due to their undeveloped heat regulation system), pregnant women (due to an elevated core body temperature during pregnancy posing risks to both the woman and the child’s health [34,35,36]), individuals with heavy physical work [37], and emergency personnel in protective clothing (e.g., firefighters) [38,39].

### 3.3. Combination Effects of Heat in Indoor Environments

#### 3.3.1. Heat and Air Quality

Increasing heat in urban areas may affect other aspects of the indoor environment in combination with heat. For example, the concentration of air pollutants can increase during a heat wave [29] and cause higher exposures in buildings without air conditioning due to infiltration (through ventilation, the building envelope, and window openings) [40]. Warmer indoor temperatures will lead to faster and increased evaporation of gas phase chemicals from building materials. As a result, these gas phase contaminants (in higher concentrations) can react with gases and particles present in the indoor air, thus forming secondary pollutants. Exposure to pollutants in indoor environments is heavily influenced by indoor air chemistry and has received increased attention, as there are health concerns [41,42].

The problems related to indoor air quality will be further exacerbated by reduced air exchange rates in naturally ventilated homes due to reduced temperature differences between the indoors and outdoors as well as keeping the windows and/or doors closed. The reduced air exchange rate results in an accumulation of pollutants in indoor air due to the decreased removal of contaminants and the allowance of a longer time for indoor air chemistry to take place and form secondary pollutants. Increased humidity combined with heat can also cause problems with condensation and mold growth [43].

#### 3.3.2. Heat and Energy Efficient Housing

Climate change may worsen existing problems in indoor environments and cause new problems. Today, buildings in most countries account for about 30–40% of carbon dioxide emissions [44,45]. Measures to reduce energy use in buildings are necessary, but if they lead to lower ventilation rates in the buildings, pollutants from indoor sources may increase in concentration, thus resulting in increased exposure [5]. Low ventilation rates increase exposure to, for example, emissions from cooking, tobacco smoke, radon, and chemical emissions from building materials [43]. There are also concerns that energy-efficient housing easily overheats due to air tightness, which reduces the possibility of eliminating heat without air conditioning [5].

There is a risk that air conditioning will be the primary means to protect the health and well-being of the population despite it leading to a dramatic increase in energy demand and, hence, greenhouse gas emissions. There is also the emerging problem that all parts of society cannot afford air conditioning to maintain cool indoor temperatures [16]. Furthermore, incidences of the sick building syndrome is higher in air-conditioned buildings than in naturally ventilated buildings [46]. In addition, increased use of air conditioning in combination with reduced ventilation rates and/or lack of regular maintenance may cause indoor air pollution to increase. On the other hand, climate change can result in a reduced need for heating and energy use during the winter causing less pollution from, for example, combustion [5].

#### 3.3.3. Heat and Social Participation

The indoor climate also has indirect effects on social participation. Lindemann et al. (2017) conducted a study in Germany on the effect of indoor temperatures on participation in social activities using the World Health Organization’s social participation score, as well as behavioral patterns such as drinking habits in the elderly during a heat wave. The study showed significant effects of high indoor temperatures on behavior and social participation. The negative relationship between high indoor temperatures and social participation was stronger if the individual had a disability or lived in a city [47].

### 3.4. Identification and Evaluation of Severe Heat in Indoor Environments

To identify severe heat stress, a combination of factors needs to be considered. A comprehensive heat stress assessment evaluates all sources of heat stress and involves three major components: the environmental factors, the physical activity (or metabolic heat production), and clothing. The environmental factors comprises four major thermal climate parameters: air temperature, radiant temperature, humidity, and air movement [48]. Thermal assessment techniques are commonly based on the body heat balance equation, by which the balance between body heat production and body heat exchange with the environment can be calculated. The human body is in heat balance when the metabolic heat production and the heat loss to the environment are equal and result in no change in the body’s heat content (S = 0). If the heat storage is positive, the body core temperature will increase and vice versa.

The body heat balance equation is as follows:

S = (M − W) − (H_res_ + E + R + C + K) [48]

S = body heat storage

M = metabolic heat production

W = external mechanical work

H_res_ = respiratory heat exchange

E = evaporative heat exchange

R = radiative heat exchange

C = convective heat exchange

K = conductive heat exchange

In epidemiological studies where the focus has been on the outdoor heat, it has been found that when the temperature rises above a certain threshold, mortality and morbidity follow suit [7,49,50,51,52]. However, similar indoor temperature thresholds have not yet been identified. Thus far, indoor conditions have been linked to thermal comfort or work in hot environments; however, these thresholds are not comparable [16]. A heat stress index often consists of a single number that integrates the effects of the fundamental factors in the human thermal environment so that variations in thermal conditions will affect its value [48]. In other words, it gives an indication of heat exposure and can, for example, be used to determine thresholds to protect public health. Many heat stress indices have been developed over the last 80 years [53], and most indexes can be categorized as rational, empirical, or direct. Rational indexes are centered on calculations that include the heat balance equation. Empirical indexes are based on the determination of equations from physiological studies of humans (e.g., sweat loss), and direct indexes are based on measurements of climate variables [45]. This section introduces five examples of workplace indices that are practical and applicable to measurements on the local level and that apply to indoor environments: The Wet Bulb Globe Temperature (WBGT) (direct), the Predicted Heat Strain (PHS) model (empirical), the Thermal Work Limit (TWL) (rational), the Equivalent Temperature (ET) (rational), and, finally, thermal comfort evaluation—in particular the Predicted Mean Vote (PMV)/PPD index (rational).

#### 3.4.1. The Wet Bulb Globe Temperature (WBGT)

The Wet Bulb Globe Temperature (WBGT) is a direct heat stress index that is easy to use and represents the thermal environment to which an individual is exposed. It is regarded as a screening method to establish the presence or absence of heat stress. The WBGT is a widespread index in indoor and outdoor occupational settings and incorporates environmental temperature, humidity, wind speed, and heat radiation (ISO 7243: 2017) [54]. The international standard for the WBGT uses a formula based on measurements of three temperature variables: T_a_, the air temperature measured with a shielded thermometer; T_g_, the globe temperature, which is the temperature inside a black globe representing the heat radiation input; and T_nw_, the natural wet bulb temperature, which is measured with a wet cloth over the sensor representing the impact of sweat evaporation on heat loss and wind speed [54]. The WBGT equation [54] for indoor environments without solar radiation is:
WBGT = 0.7 T_nw_ + 0.3 T_g_

The index can be adjusted to account for clothing and the metabolic heat production is estimated from a time-weighted mean value based on a reference table [54]. Table 1 below shows the WBGT limit values for acclimatized and unacclimatized people for five classes of metabolic rate.

#### 3.4.2. The Predicted Heat Strain (PHS) Model

The Predicted Heat Strain (PHS) model (ISO 7933) is one of the most common indexes for the evaluation of thermal environments and related physiological strain based on the heat balance equation. It is also an international standard [55,56,57]. The model calculates the time when the core body temperature and sweat loss reach critical levels and includes all factors that affect human heat exchange with the surrounding environment. The model is based on an analysis of human heat exchange and sweat rate for the maintenance of a stable core body temperature, allowing for predictions of exposure limits in relation to dehydration (5% of the body mass for 95% of the working population or 7.5% for an average subject) and core temperature (38 °C) [57,58].

#### 3.4.3. The Thermal Work Limit (TWL)

The Thermal Work Limit (TWL) is a rational heat stress index and is designed for self-paced workers. It puts limits on the metabolic rate (in W/m^2^) that a euhydrated, acclimatized individual can maintain in a specific thermal environment within a safe body core temperature (<38.2 °C) and sweat rate (<1.2 kg/h^−1^). The index also accounts for clothing, and the assessment can be extended to unacclimatized persons. The index is based on an iteration of equations to provide the heat flow from “skin to environment” due to convection, radiation, respiration, skin wettedness, maximum evaporation rate and efficiency of sweating, heat flow from “core to skin”, heat flow due to respiration, and sweat rate. There will then be a unique mean skin temperature and metabolic rate that provides a heat balance, at which point the body is in thermal equilibrium with the environment. The index is not as easy to use as the WBGT but allows not only thermal strain to be evaluated but also the impact of various strategies such as improved local ventilation or refrigeration to be quantitatively assessed [53].

#### 3.4.4. The Equivalent Temperature (ET)

The Equivalent Temperature (ET) is a rational thermal comfort index. It includes air temperature, radiant temperature, and air velocity in its calculation of dry heat loss [59]. It is easy to use with a ‘thermal comfort’ instrument that directly calculates the ET and consists of a heated, ellipsoid sensor representing the physical model of the human body (corresponding to the total convective and radiative heat exchanges). Basic climatic parameters can also be used to calculate the index, i.e., air and mean radiant temperature and air velocity. The index provides a measure of the diversion from thermal neutrality [60]. The index is rarely used today as a comfort index but has been applied to assess heat stress [48].

#### 3.4.5. Thermal Comfort Evaluation

It is important for public health to maintain thermal comfort in indoor environments [61]. Commonly, people are comfortable within a narrow range of temperatures and according to the WHO, people on average do not perceive comfort in temperatures outside the range of 17–31 °C [61]. However, the tolerance range of an individual is usually less than this and tends to be even narrower with age or disability [61]. Thermal comfort is strongly linked to health and well-being. The WHO has therefore made the general assumption that the thermal comfort range is 18–24 °C, which is an interval specified in current comfort standards [28,61]. Indoor comfort is affected not only by the climate, season, acclimatization of individuals, and thermal history but also economic and cultural contexts and expectations. A warmer climate may therefore create a higher limit on comfort, such as 28 °C in Greece and 25 °C in France [62]. Generally, there are three assessment methods of thermal comfort:A psychological assessment that sees thermal comfort as a state of mind that expresses satisfaction with the thermal environment [9].A thermophysiological assessment focusing on the thermal receptors in the skin and hypothalamus and based on the heat balance equation [9].An adaptive assessment based on field studies that look at adaptive comfort, psychology, and behavior. The assessment takes into consideration the local context, for example, the indoor comfort temperature is calculated from the outside temperature using an adaptive algorithm [63].

On the international level, standards for thermal comfort have been developed by several standards organizations; for example, the International Standardization Organization (ISO), the European Standardization Organization (CEN), and the American Society of Heating, Refrigeration and Air Conditioning Engineers (ASHRAE). However, the standards have been criticized because most standards have been developed by experts from Europe, North America, and Japan that focus on the heating/winter season and therefore do not apply to warmer climates [64]. The International Standard for Thermal Comfort, ISO 7730, uses a formula named the Predicted Mean Vote (PMV). PMV predicts a numerical value of the mean subjective response to the perceived thermal comfort/sensation based on knowledge of the six thermal factors [65].

Scales for estimating thermal comfort have also been developed to evaluate the thermal comfort of an individual, such as the ASHRAE scale and ISO 10551 *see the example in Figure 1) [65,66].

However, several researchers are critical of this assessment model and argue that the model was developed with a mechanized system in mind. As an alternative, they propose a more adaptive model [62,63,67,68,69]. In the PMV strategy, it has been assumed that air conditioning improves comfort and productivity, and that all people around the world have similar comfort needs [69]. Consequently, traditional buildings that were considered comfortable by previous generations can often not meet current standards [62]. Adaptive comfort studies have shown the diversity of the environments that people think is comfortable, which cannot be explained by physiological models [62]. Field studies of thermal comfort demonstrate that personal control of the indoor climate through, for example, adjustable thermostats, opening windows, individualized ventilation, and other controls, have a positive impact on comfort and productivity and prevent the sick building syndrome [70]. Related to this criticism, DeDear and Brager (2002) proposed a new thermal comfort standard for naturally ventilated buildings, which means that the ISO 7730 and ASHRAE standard should only be used in heated and air-conditioned buildings. The approach in the new thermal comfort standard helps especially architects who want to apply low energy solutions [67].

#### 3.4.6. The Need for the Development of a Heat Index for Indoor Environments

In order to establish indoor heat stress thresholds, it is necessary to develop an index or modify existing indices to reflect heat effects on health of indoor occupants. All the indices introduced—namely EBGT, PHS, TWL, ET and PMV/PPD—have been developed for healthy working populations, and a heat index for indoor environments needs to cover many different aspects such as vulnerable groups and different uses (e.g., retirement homes, preschools, and offices). It also needs to include daily variations as well as temperature changes between days, as research has shown that high nighttime temperatures can adversely affect health due to lack of recovery [12,15].

This section will continue to give four example suggestions of indices and models that could be used, developed, and built upon. The first suggestion was developed by Chan et al. (2001), who developed a physiologically based model to assess the most important health-related risks called the Heat-Related Health Effects Index (HEI). The HEI index is flexible and contains site-based assessments of health effects [25].

The second suggestion was developed by the German Meteorological Institute (Deutscher Wetterdienst, DWD), who has expanded the existing Heat-Health Warning System (HHWS) based on the Perceived Temperature thermal index [71] with a thermal building simulation model that estimates the indoor heat load [72] and calculates heat stress based on the ISO 7730 PMV/PPD model [73]. However, the building simulation model is limited to the worst-case scenario for indoor conditions and estimated by air temperature only [74]. Below is a figure of the model database set-up (Figure 2).

The third suggestion was developed by Walikewitz et al. (2018), who applied the Universal Thermal Climate Index (UTCI) in indoor environments in Berlin in the summer of 2013 and 2014. The authors found that all rooms reached the heat stress threshold according to UTCI levels, especially during heat waves. UTCI levels showed great variations within the city and within a building. Heat stress occurred below 35% of all days during both night and day. By comparing the thermal load between night and day, the maximum threshold was identified as a UTCI of 32 °C for severe heat stress at night [15].

The fourth suggestion was developed by Smargiassi et al. (2008), who applied a mapping tool using a Geographic Information System (GIS) model for urban indoor temperatures in time and space for all residential buildings in Montreal. A general estimation equation model (GEE) was developed to predict indoor temperatures that integrated variations in outdoor temperatures with georeferenced determinants available for the entire city such as surface temperatures at each location (from a satellite image) and building characteristics (from the Montreal Property Assessment database). The model found that the indoor temperature may increase by up to 54% in °C compared to the outdoor temperature [75].

### 3.5. Solutions to the Development of Severe Heat Stress in Indoor Environments

#### 3.5.1. Microclimate Solutions

Primarily, it is important during a heat wave that people are conscious, pay attention to symptoms of heat stress, and follow advice. Based on the reviewed articles, on an individual level, there are several actions that can be taken through behavioral, physiological, and psychological adjustments. Below in Table 2 are descriptions and example actions of the three adaptation levels:

One example of an effective behavioral action is the cooling of extremities. Research has shown that when extremities are cooled, the whole body’s heat load is reduced [79]. The cooling of hands or feet, when utilizing arterio-venous anastomoses (AVAs), has proven to be an effective measure for people exposed to high heat loads (e.g., during military activities and firefighting). AVAs consist of the direct connection between small arteries and small veins. There are many AVAs in hands and feet and they play an important role in temperature regulation in humans [80]. In hot environments, heat stress is reduced significantly if hands and feet are cooled through cool water immersion [81,82], and the method is therefore especially applicable in acute situations for rapid cooling of the body.

The rest of this section will focus on technical measures at the individual level, with a particular focus on personal cooling systems that have received great attention in research publications in recent years. It is recognized that personal cooling can improve both thermal comfort and save energy. Personalized cooling solutions include shade structures, water-based cooling, smart textiles, ventilated clothing, personal ventilation, personal humidifiers, fans, air conditioning, and cooling clothes (especially vests) using air or liquids [37,83,84,85]. When using ventilation solutions inside clothing, the circulating air speeds must be around 100 L/min in order to have a significant effect. At air speeds below 30 L/min, the effect is hardly noticeable but also depends on the actual ambient conditions and system characteristics [86]. In the most effective personal cooling systems, the cooling media covers large body areas to maximize heat transfer—in ventilated clothes, for example—or are placed in areas where blood vessels near the skin surface are abundant or near specific areas that need to be cooled (contact cooling). These areas include the neck, wrists, and forearms, but the cooling media should at the same time not limit heat dissipation.

In evaluating personal cooling systems, most studies have so far focused on laboratory experiments and studies of office environments. For example, He et al. (2017) evaluated different personal cooling systems in an office environment and found that desktop fans were very energy efficient and that, in combination with a reflective cooling desk, extended the employee’s thermal comfort range [87]. However, desk fans fit best in hot and dry environments if the air temperature is not much above 40 °C [88,89,90]. At any humidity level, increased airflow is always beneficial at 34 °C air temperature or below with minimal clothing (the author’s own unpublished data).

A wearable personal passive cooling system that enhances mobility has received increasing attention: The use of Phase Change Materials (PCM) in vests and clothing. PCM is a heat storage material, such as ice, frozen gel, salt, or wax, which has the inherent property of absorbing or releasing heat as they change phase, e.g., from solid to liquid (melting) and back to solid (crystallization). Therefore, a PCM material has two types of thermal effects: A cooling effect when it melts and a warming effect when it solidifies. Gao et al. (2012) found that subjects experienced an improvement when using a vest integrated with PCM in a warm climate. The results indicate that personal cooling with PCM can be used as an alternative to improve thermal comfort without the use of air conditioning and may also be applicable to vulnerable groups. However, specific environmental conditions affect the effect, and PCM materials would therefore fit best in a hot humid climate or under protective layers that do not allow good sweat evaporation [91].

Personal cooling also contributes to energy savings because the energy is only used where it is needed [84,91]. Vesely et al. (2014) showed that energy savings of up to 60% could be achieved with personal air conditioning [84]. Pan et. al. (2005) evaluated the performance of a personal air conditioning system compared with a central system. It was found that the thermal comfort index PMV (ISO 7730) was always lower for the personal system compared with the central system. The authors also found that the personal system could save up to 45% of the energy use compared to that consumed by the central system [92]. However, personal air conditioning systems are best suited to environments where sweat evaporation is limited—for example in hot and humid environments [93].

In dry conditions, personal cooling by the use of a humidifier facing the body may in some cases provide effective cooling through evaporation. However, the systems could have the disadvantage of forming a favorable environment for mold and bacteria [83]. Chakroun et al. (2011) reported that the use of personal humidifiers against the body’s upper body segments, especially the face, was sufficient to achieve comfort [83]. El Hourani et al. (2014) found that personal humidifiers were an effective system for implementation in Lebanese office environments [94].

#### 3.5.2. Macroclimatic Solutions

Based on the reviewed articles, this section focuses on climate-sensitive urban planning and building design, building characteristics and links to the build-up of severe heat levels indoors, innovative and low-energy cooling technologies, and finally, the use of PCM material as a passive cooling technology.

An unfavorable indoor climate can be prevented or mitigated against by applying the principles of climate-sensitive urban planning and building design that are adapted to the residential area, urban, region, and climate [95]. This includes traditional methods for creating a comfortable indoor climate and measures that make use of differences between day and night conditions [68] that utilize high thermal mass in the building material and cooling by means of evaporation of water [10]. Climate-sensitive building design consists of strategies to maximize ventilation and minimize heat from solar radiation, incorporating aspects such as orientation of the building, number, size and location of windows or glass walls, properties of building materials, use of shading and reflective materials, and paints [16,28,96]. The structure of a building is important because a heavy foundation has high heat capacity and can smooth the temperature variations and lower indoor temperatures. The orientation of streets and buildings in relation to the prevailing wind direction also has a major impact on both outdoor and indoor ventilation [10,97]. Overhang shading, either through vegetation or shading devices, is crucial for creating a comfortable indoor climate [98,99]. Cold climate countries, like Scandinavia, traditionally have buildings that are insulated against the cold, which is also beneficial for the protection against outdoor heat. However, modern architecture tends to design buildings with large south-facing windows using lightweight building materials or glass, which is problematic during hot weather [100].

In regards to building characteristics, studies have shown that upper floor apartments with non-opening windows are associated with increased mortality [14,101,102]. White-Newsome et al. (2012) found that the indoor temperature of single-family houses constructed of vinyl panels or wood were more sensitive to changes in the outdoor climate than brick houses. The results indicate that brick buildings help to protect from outdoor heat [14]. Roaf et al. (2009) showed that the main causes of mortality indoors during the heat wave in Paris in the summer of 2003 were the lack of insulation in the building and living on the upper floor apartments. The risk was lower when the homes had more rooms (and possibilities for creating drafts) and larger when there were more windows (increased heat load from solar radiation). The orientation of the building, the possibility to create draft, surface temperatures, and vegetation were other factors that affected the indoor climate [18]. Solar radiation through windows is the factor that influences the cooling needs the most. It is therefore important to reduce solar radiation by using, for example, sun-reflecting window glass [103] and install shading devices, such as external or interior blinds. External shading devices have been shown to be more effective than interior blinds [104]. Furthermore, the material of the shading device matter. Alawadi et al. (2012) investigated solar thermal load in buildings with windows having blinds with a built-in PCM material. The result indicate that a PCM with a high melting temperature has the best thermal performance and that the heat increase through windows can potentially be reduced by one third [105].

Research and development of innovative and low energy consuming cooling technologies and strategies is ongoing [106,107,108,109]. District cooling and air conditioning powered by renewable energy are potential alternatives to conventional air conditioning [110]. One example of the application of district cooling is in Singapore, however, it is not a common solution globally [111]. Trygg and Amiri (2007) found that the conditions in Sweden are favorable for converting district heating to district cooling, although its application has so far been limited [112]. Passive and active solar air conditioning systems have great potential to replace conventional cooling technology [113,114]. Solar cooling has attracted much attention, as cooling needs often coincide with strong solar radiation. Allouhi et al. (2015) provide a good overview of what types of available technologies use both passive and active alternatives and show significant energy savings for European conditions. Furthermore, these systems have the advantage of contributing to heating and hot water production during periods when cooling is not required [109].

Passive cooling techniques are also promising alternatives to high energy consuming air conditioning. Of the different passive cooling strategies, the use of PCMis an effective way of increasing the thermal resistance of the building envelope with its inherent heat energy storage. It also reduces temperature fluctuations. The integration of PCM into building materials such as walls, floors, and ceilings [115] has been investigated as a potential technique for reducing the cooling needs of buildings. PCMcan be classified into three main groups: Organic, inorganic, and eutectics. Akeiber et al. (2016) found that the organic type received the most attention due to its reasonable price, stability, non-corrosivity, and high fusion. Akeiber et al. (2016) provide an updated compilation of PCM options and its possible integration into buildings [45]. Furthermore, results from Silva et al. (2016) show great potential of PCM to improve the thermal performance of a building, especially through its integration into glass and shading devices [116]. PCM technology applied to the building’s façade can also significantly improve thermal performance [116]. Medina et al. (2008) found that wall panels integrated with a concentration of 10% PCM reduced heat intrusion by an average of 37% during the summer. A PCM concentration of 20% reduced the heat intrusion by an average of 62%. The authors also confirmed that the higher the temperature difference between day and night, the better the performance of the PCM material [117]. In other words, countries with large temperature differences between day and night, such as Scandinavian countries, can benefit greatly from the integration of PCM into building materials. Finally, Aranda-Usón et al. (2014) show that the use of PCM in addition to saving energy also reduces the overall environmental impact of a building. However, this decrease varies greatly with the climatic conditions and the type of PCM used [118]. This result has been backed up by life cycle analyses [119]. Summing up, the use of PCM is promising; however, it is important that the environmental and health impacts of the PCM are assessed before it is introduced into a building.

## 4. Conclusions

Increasing heat in urban areas affects buildings and their indoor environments. However, few studies have focused on the thermal environment indoors during heat waves, although people spend most of their time indoors. Most Heat-Health Warning Systems (HHWS) are also based on outdoor conditions, although people are likely to experience increased heat indoors.

The heat indoors differs significantly from the outdoors, where the literature review found that the indoor temperature could increase by up to 50% more in °C compared to the outdoor temperature. There is a link between the outdoor and indoor climate in non-air-conditioned buildings; however, the connection is complex and consists of interactions between natural processes with technology, type of building, individual behavior, and social systems.

Exposure to heat have negative health outcomes, and, in order to identify severe heat, six factors need to be considered: air temperature, heat radiation, humidity, air movement, clothing, and the physical activity of the individual. Severe heat can be identified using a heat index that includes the six factors. The paper highlights some examples of practical and easy to use heat indices relevant to indoor environments: The Wet Bulb Globe Temperature (WBGT), the Predicted Heat Strain (PHS) model, the Thermal Work Limit (TWL), the Equivalent Temperature (ET), and the thermal comfort index PMV/PPD. However, all indices are limited as they were developed for healthy working adults. In order to determine severe indoor heat relevant for public health, it is necessary to develop a heat index or adjust existing thresholds to include vulnerable groups, different uses and daily variations.

There are several effective local level actions that can be taken to reduce indoor heat and improve the health and well-being of the urban population. These include shading devices, personal cooling techniques such as cooling desks, cooling vests and fans, ventilation, and integration of Phase Change Materials (PCM) into facades, ceilings, floors and windows. Air conditioning based on renewable energy, such as passive and active solar energy, is also a promising solution. Finally, communication efforts focusing on the effectiveness of measures for different climates and information about these solutions at correct time points is of utmost importance.

## Figures and Tables

**Figure 1 ijerph-16-00560-f001:**
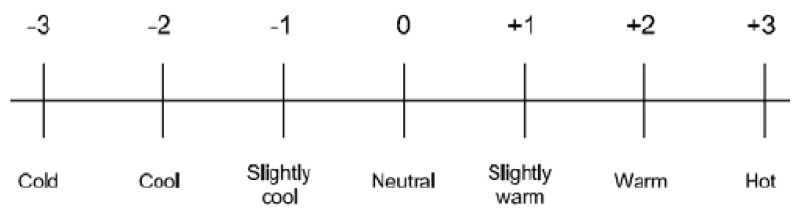
American Society of Heating, Refrigeration and Air Conditioning Engineers (ASHRAE): s 7-point scale for thermal comfort [65].

**Figure 2 ijerph-16-00560-f002:**
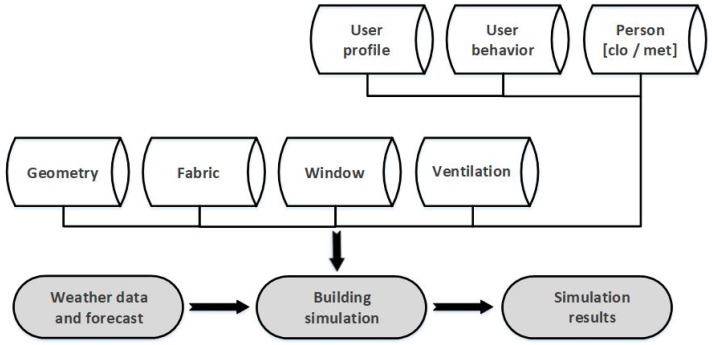
Database set-up of the German model who extended the current Heat-Health Warning System (HHWS) with a thermal building simulation model, modified on the basis of [73].

**Table 1 ijerph-16-00560-t001:** Recommended Wet Bulb Globe Temperature (WBGT)_eff_ reference values for acclimatized and unacclimatized people for five classes of metabolic rate, modified on the basis of [54].

Metabolic Rate (Class)	Metabolic Rate (W)	WBGT Reference Limit for Person Acclimatized to Heat (°C)	WBGT Reference Limit for Person Unacclimatized to Heat (°C)
Class 0 Resting metabolic rate	115	33	32
Class 1 Low metabolic rate	180	30	29
Class 2 Moderate metabolic rate	300	28	26
Class 3 High metabolic rate	415	26	23
Class 4 Very high metabolic rate	520	25	20

**Table 2 ijerph-16-00560-t002:** Examples of actions at different adaptation levels of the individual.

Adaptation Level	Example Actions
**Behavioral**	adjusting clothes and body movement such as posture and activity; - adjusting conditions including opening or closing of windows; - creating ventilation/draught; - taking a cold shower; - applying water on face/neck, hands or using wet towels; - adjusting the thermostat; - installing ceiling and table fans; - taking a siesta during the hottest hours of the day; - customizing cooking by the day and, for example, not using oven; - change blinds to block unwanted sunlight; - move between rooms (i.e. from kitchen to cooler areas such as a basement) [76,77]; - using ventilation solutions in clothing (described further in text below); - using personalized table fan cooling (described in the text below);
**Physiological**	- conduct regular physical exercise; - adopt a systematic heat exposure regime to induce acclimatization [48].
**Psychological**	- create awareness of the thermal environment. Here, perceptions can be influenced by the individual’s previous thermal experiences and thus expectations of the building in which the person accommodates [78].

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
