# Peer review of "Heat Stress in Indoor Environments of Scandinavian Urban Areas: A Literature Review"

_ijerph, 2019, doi:10.3390/ijerph16040560_

Reviewer 1 Report

Lundgren Kownacki et al give a comprehensive overview of the current state of the art of
heat stress in the indoor environment. The manuscript is well written and I recommend the
paper for publication after the following issues have been addressed:

Title: I have the impression that most of the cited literature does not consider the
indoor environment in Scandinavia. Therefore, I suggest to remove the term
'Scandinavian' from the title.

L19: This is wrong. At least one HHWS (Germany) includes indoor temperatures in their
warning system as you state in L344ff. The approach has been described in Pfafferott and
Becker (2008). See also Matzarakis 2017 for the most recent description of the German
HHWS.

Chapter 3.4.1-3.4.4:
I do not understand how the four indices were chosen. Why is PET or UTCI not appropriate
for indoor use? I recommend Staiger et al (2019) for a comparison of appropriate thermal
indices.

L 298: How was the range 17-31°C defined? Please give a reference

L 305: 'standadrs' -> standards

L344ff:  The best reference for the building simulation model in the German HHWS is
Pfafferott and Becker 2008. Unfortunately, the paper is written in German, the abstract,
however, is also available in English. Heat stress in the building simulation module is
estimated by air temperature only.

L361: a temperature increase by 54% does not make much sense, since 54% differs
substantially depending on the unit of the temperature measurement (K, °C, °F)
This also applies to L507  and L18.

References

Pfafferott, Jens, und Paul Becker. „Erweiterung des Hitzewarnsystems um die Vorhersage
der Wärmebelastung in Innenräumen“. Bauphysik 30, Nr. 4 (2008): 237–43.
https://doi.org/10.1002/bapi.200810031.

Matzarakis, Andreas. „The Heat Health Warning System of DWD - Concept and Lessons
Learned“. In Perspectives in Atmospheric Sciences, Editors: Karacostas et al,
191–96. Springer Atmospheric Sciences, 2017.

Staiger, Henning, Gudrun Laschewski, und Andreas Matzarakis. „Selection of Appropriate
Thermal Indices for Applications in Human Biometeorological Studies“, 2019, 15.

Author Response

Lundgren Kownacki et al give a comprehensive overview of the current state of the art of
heat stress in the indoor environment. The manuscript is well written and I recommend the
paper for publication after the following issues have been addressed:

Title: I have the impression that most of the cited literature does not consider the
indoor environment in Scandinavia. Therefore, I suggest to remove the term
'Scandinavian' from the title.

Response 1: Dear reviewer. First of all, thank you for taking your time to review our paper and for the constructive comments and suggested literature. Regarding the removal of ‘Scandinavian’ in the title, we have decided to keep it despite the lack of research papers looking into Scandinavian indoor environments. This was the original aim and motivation for the review despite the lack of research. We now have added more Scandinavian literature in relation to the climatological aspects of heat waves in the introduction. We hope that adds more justification to keep ‘Scandinavian’ in the title.

L19: This is wrong. At least one HHWS (Germany) includes indoor temperatures in their
warning system as you state in L344ff. The approach has been described in Pfafferott and
Becker (2008). See also Matzarakis 2017 for the most recent description of the German
HHWS.

Response 2: We agree. To address this we have changed ‘current’ to ‘most’ HHWS. I hope this is satisfying. Also, thank you for the suggested description. We have read and added Matzarakis to the reference list.

Chapter 3.4.1-3.4.4:
I do not understand how the four indices were chosen. Why is PET or UTCI not appropriate
for indoor use? I recommend Staiger et al (2019) for a comparison of appropriate thermal
indices.

Response 3: The motivation to the choice of thermal indices has now been modified (we have added ‘local level’):

This section introduces five examples of workplace indices that are practical and applicable to measurements on the local level and that apply to indoor environments

We have selected the indices based in their practicality for use directly by measuring on the local scale. Steiger et al. reviews relevant biometeorological indices for use in urban HHWS. I hope this is now clearer to you.

L 298: How was the range 17-31°C defined? Please give a reference

Response 4: It’s from a World Health Organization report on the health impact of low indoor temperatures. The reference has been added and the sentence modified to include the source.

L 305: 'standadrs' -> standards

Response 5: Thank you. It has been changed accordingly.

L344ff:  The best reference for the building simulation model in the German HHWS is
Pfafferott and Becker 2008. Unfortunately, the paper is written in German, the abstract,
however, is also available in English. Heat stress in the building simulation module is
estimated by air temperature only.

Response 6: Thank you for this input. The reference and a sentence has been added to the manuscript. Please refer to the text.

L361: a temperature increase by 54% does not make much sense, since 54% differs
substantially depending on the unit of the temperature measurement (K, °C, °F)
This also applies to L507  and L18.

Response 7: Thank you for this valuable comment. Of course that is the case. We have inserted the unit °C to specify in each case. Hope you find this sufficient.

References

Pfafferott, Jens, und Paul Becker. „Erweiterung des Hitzewarnsystems um die Vorhersage
der Wärmebelastung in Innenräumen“. Bauphysik 30, Nr. 4 (2008): 237–43.
https://doi.org/10.1002/bapi.200810031.

Matzarakis, Andreas. „The Heat Health Warning System of DWD - Concept and Lessons
Learned“. In Perspectives in Atmospheric Sciences, Editors: Karacostas et al,
191–96. Springer Atmospheric Sciences, 2017.

Staiger, Henning, Gudrun Laschewski, und Andreas Matzarakis. „Selection of Appropriate
Thermal Indices for Applications in Human Biometeorological Studies“, 2019, 15.

Reviewer 2 Report

Dear Authors,

The paper corresponds with the current research trend in climatology. I believe the article is suitable for publication, although it requires serious changes.

Introduction

The introduction requires a supplementation of papers analysing multiannual changes in the occurrence of heat waves in Scandinavian. The issue is popular in climatological literature, in particular in recent years. This chapter should be expanded by discussing the results with other documents on thermal conditions in the city during heat waves.

The proposed literature:

Hans Orru and Daniel Oudin Åström 2017. Increases in external cause mortality due to high and low temperatures: evidence from northeastern Europe. International Journal of Biometeorology 61, 5, 963–966.

Arkadiusz M. Tomczyk et al. 2017.  Warm spells in Northern Europe in relation to atmospheric circulation. Theoretical and Applied Climatology 128 (3-4), 623-634.

Daniel Oudin Åström et al. 2018. Heat wave–related mortality in Sweden: A case-crossover study investigating effect modification by neighbourhood deprivation. Scandinavian Journal of Public Health.

Marek Półrolniczak et al. 2018. Thermal Conditions in the City of Poznań (Poland) during Selected Heat Waves. Atmosphere 9, 11.

Author Response

Dear Authors,

The paper corresponds with the current research trend in climatology. I believe the article is suitable for publication, although it requires serious changes.

Introduction

The introduction requires a supplementation of papers analysing multiannual changes in the occurrence of heat waves in Scandinavian. The issue is popular in climatological literature, in particular in recent years. This chapter should be expanded by discussing the results with other documents on thermal conditions in the city during heat waves.

Response 1: Dear reviewer. First, thank you for taking your time to review our paper and for the constructive comments and suggested literature. As our literature search focused on the indoor environment, these climatological studies on the outdoor temperature and mortality relationship did not show. We have now added a selection of your suggested literature relevant for the Scandinavian context in the introduction.

The proposed literature:

Hans Orru and Daniel Oudin Åström 2017. Increases in external cause mortality due to high and low temperatures: evidence from northeastern Europe. International Journal of Biometeorology 61, 5, 963–966.

Arkadiusz M. Tomczyk et al. 2017.  Warm spells in Northern Europe in relation to atmospheric circulation. Theoretical and Applied Climatology 128 (3-4), 623-634.

Daniel Oudin Åström et al. 2018. Heat wave–related mortality in Sweden: A case-crossover study investigating effect modification by neighbourhood deprivation. Scandinavian Journal of Public Health.

Marek Półrolniczak et al. 2018. Thermal Conditions in the City of Poznań (Poland) during Selected Heat Waves. Atmosphere 9, 11.

Round  2

Reviewer 1 Report

no futher comments, thank you.